# The nose knows: Thermal responses to active psychological stressors

**Perrine Theroude**[1], **Marianne Paisley**[1], **Matthew Thompson**[1‡], **Amelie Wheeler**[1‡], **Georgina Donati**[2¤], **Gillian S. Forrester**[1*]

**1** School of Psychology, University of Sussex, Brighton, United Kingdom, **2** Department of Psychiatry, Warneford Hospital, University of Oxford, Oxford, United Kingdom

☉ These authors contributed equally to this work
‡ These authors contributed equally to this work
¤ Centre for Brain and Cognitive Development, Birkbeck College, London
* g.forrester@sussex.ac.uk

## Abstract

Stress is an essential component of our lives. It helps to us to keep alert, stay motivated, and adapt to new and challenging situations. However, it is also a leading cause of poor mental wellbeing. Investigating psychological stress is essential to improving both the physical and mental health of the general population. Current methods often rely on self-report and physically invasive (contact) measures which can lack objectivity and ecological validity. Thermal imaging is emerging as a powerful objective, continuous, and physically non-invasive (non-contact) tool to investigate psychological stress through changes in nasal skin temperature. Yet there remain gaps in our understanding of thermal ranges, thermal recovery, and thermal associations with perceived stress in the healthy population. We present a new protocol, employing continuous thermal video to measure nasal temperature fluctuations during stress induction in healthy adults. Results indicate that induced psychological stress significantly decreases nasal temperature compared with a white noise baseline, and a social stress task elicited a significantly stronger nasal temperature decrease from baseline compared with a cognitive stress task. Although perceived stress was not associated with nasal thermal fluctuations, perceived somatic anxiety symptoms did significantly relate with nasal temperature change. These findings reveal new insight into the psychological and physiological human stress experience. The continuous, non-contact and objective benefits of thermal imaging makes it uniquely placed to contribute to real-world health applications, including translation to clinical and nonverbal populations across the lifespan.

**Data availability statement:** Data are available on the Open Science Framework: DOI: https://osf.io/xb8w5.

**Funding:** The PI (Forrester) received a 'Discipline Hopping' grant from the Natural Environment Research Council (NE/X018245/1) to develop a thermal imaging protocol for great apes. This protocol informed and inspired the current experimental work with humans. The funder did not play any role in the study design, data collection and analysis, decision to publish or preparation with the manuscript. Funder URL: https://www.ukri.org/councils/nerc/.

**Competing interests:** The authors have declared that no competing interests exist.

## Introduction

Acute stress is a necessary human experience that arises when we are threatened or perceive ourselves to be threatened [1]. It is relatively short-term, elicited by stimuli with a discernible start and end. Whether something is stressful is first judged via sensory processing and can include stored memories of previous stressful situations [2]. Stressors can be any factor that causes physiological or psychological strain, or both. Anytime we are exposed to a stressor the nervous system is activated as part of an evolutionary adaptive mechanism to prepare an organism to make the appropriate behavioural response (e.g., fight, flight) [3]. However, once a threat has passed, the body strives to return to a balanced state (homeostasis). In humans, there is considerable variation in how stress is experienced because it is the synthesis of both physiological arousal and psychological interpretation of the physiological arousal. Because stress plays a significant role in our wellbeing [1], we require a comprehensive understanding of how physiological and psychological stress are associated.

Investigations of stress often use self-report measures (e.g., surveys) to evaluate the intensity of an individual's stress experience, informing factors such as valence. These measures are interesting and can capture relatively stable characteristics of how individuals perceive stressful situations in their lives [4]. For example, individuals may rate how stressful situations impact their lives (over a prescribed timeframe, e.g., Perceived Stress Scale; [5]), or how specific stressors impacts one's perceived cognitive and physical stress (e.g., State Trait Inventory for Cognitive and Somatic Anxiety; [6]). However, self-reports possess limitations because they rely on what people 'believe' about themselves, using language-based descriptions and children, and adults alike, are not always good at recognising their own stress levels [7]. To further complicate matters, the stress experience is influenced through social and cultural contexts [8–9], and not all individuals can tell us how they feel, including infants, children, and individuals with nonverbal conditions. Taken together, the factors influencing perceived stress are wide and varied, complicating population comparisons.

Physiological stress, on the other hand, can be evaluated using more objective measures elicited by activation of the sympathetic nervous system as it prepares the body to react to threat (e.g., increased heart rate, excessive sweating) [3]. Some physiological stress measures associate with self-report psychological stress survey scores like facial muscle activity, skin conductance, heart rate, and brain waves [10–12]. However, capturing these more objective stress measures also has limitations. While considered largely non-invasive, physiological stress measures are obtrusive, requiring physical contact with participants and disruption of naturalistic behaviour. In some cases, the sole act of conducting contact physiological readings can cause a spike in stress levels (e.g., White Coat Hypertension; [13]) that is not associated with the experimental manipulations. Additionally, many contact measures provide snapshots of the stress response rather than continuous sampling (e.g., blood pressure, respiration). Moreover, links between perceived stress and physiological stress that are common in the laboratory do not always translate to research using real-life scenarios, calling into question the ecological validity of these

measures. For example, while perceived stress and heart rate variability are commonly associated in the lab setting, only a small relationship between the two variables is maintained outside of a laboratory setting, suggesting that the sensitivity of some contact measures might be specific to specific stimuli and may not possess adequate sensitivity to detect stress in more ecologically valid settings [14]. To obtain a comprehensive understanding of how physiological and psychological stress associate, we require objective, continuous and non-contact measures that can be applied across different populations, settings and even other species.

Although infrared thermography (IRT) was designed and manufactured as an industry-based tool for detecting heat leaks and for surveillance in airports, it is emerging as a powerful non-contact translational tool in medicine for revealing infection [15–17], inflammation [18], and tumours [19]. It is also showing significant potential to contribute to the study of stress and arousal [3,20–24]. Humans are homeothermic, meaning we maintain a stable internal body temperature; fluctuations in our temperature associate with activation of our sympathetic nervous system, as it drives blood flow through vasoconstriction and vasodilation [25], which can be captured by IRT [26]. The changes in blood flow direction around the face through vasoconstriction and vasodilation can be measured as local decreases and increases in skin temperature respectively [27]. An increase in temperature is elicited by a rise in blood perfusion, whereas a decrease in temperature is the result of decreased facial irrigation [20]. IRT has successfully demonstrated robust changes in facial skin temperature influenced by changes in blood flow [28–30] using analyses that consider facial regions of interest (ROIs) [31]. Facial temperature changes based on vasomotor activity have been localised to the forehead, nose, upper lip, and cheeks [28–30,32–34].

Studies show that facial temperature can be influenced by ambient temperature (e.g., cooling and warming) [35–36] and by psychological stress [37]. Researchers have been successful in capturing psychologically induced stress using IRT, whilst controlling for the ambient temperature of the environment [17,38–41]. Stress induced by using fear stimuli has been most widely investigated with IRT [42]. While a range of human facial ROIs are influenced by fear stimuli (e.g., nose, mouth, cheeks, forehead, periorbital and maxillary regions), research indicates that the nose tip is the most sensitive to negative stressors, particularly associated with fear [42].

Thermal changes in the tip of the nose have been used repeatedly to estimate physiological stress responses because the nose offers a facial region with no underlying muscles, controlling for thermal changes due to muscle contraction of facial expressions, but while still being affected by the sympathetic nervous response, vasoconstriction [20,33,43,44]. The nasal dip also correlates well with other physiological signals that are known to be impacted by stress activation. For example, McDuff et al. 2023 [45] demonstrated with 85% accuracy, with IRT, the classification of stress or no-stress conditions when considering heart rate, breathing rate and heart rate variability, supporting the concept that stress-induced decreases in the nasal thermal temperature are directly associated with autonomic nervous system-driven vasomotor activity.

Studies with adults have consistently demonstrated a decrease in the nasal temperature associated with psychological stress [17,21,46,47] whilst controlling for the ambient temperature of the environment [17,38–41]. Fewer studies have been conducted using IRT on children, but these also report a nasal dip in temperature associated with physiological stress [48–49]. The nasal decrease in temperature has been reported in children aged 7–11 [3], in children as young as 38–42 months during an investigation of mild distress [34,43], in children experiencing mild posttraumatic stress disorder [42] and in both mothers and their baby when the dyads are evaluated under a stress condition [50–51]. Based on the literature to date, the nasal region is emerging as a consistent physiological indicator of stress.

Although IRT provides a powerful non-contact and objective measure of stress, there still exist some limitations. Most studies capture thermal readings while participants passively react to audio and visual stressors (e.g., fear expression) as opposed to actively behaving in stressor conditions, inhibiting an understanding of stress responses 'in situ'. IRT studies generally employ a relatively low sampling rate, decreasing our understanding of the timing of the stress response and recovery from the stressor. For example, thermal temperatures are often collected using static images taken every 3–60

seconds [43,52–54] limiting the detail in temperature change onsets, fluctuations, and recovery to baseline. Advances in thermal technology offers a multitude of thermal video camera options with good resolution for experimental research purposes allowing for both higher-frequency data sampling and flexibility to acquire more ecologically valid datasets.

Studies with primates have taken a more ecologically valid approach to investigating stress with IRT, and the nasal dip has also been reported in both monkeys and great apes. In laboratory settings, monkeys demonstrated a nasal dip when introduced to settings with a threatening person [54] and when exposed to negative audio and/or visual stimuli [28]. In chimpanzees, a mean nasal dip of 1.5°C occurred within 2-minutes of exposure to a negative stressor (auditory playback of fighting conspecifics) and recovery back to baseline occurred within the same timeframe [55]. Studies of wild apes indicate robust transfer of the nasal dip to naturalistic behaviour outside of the laboratory. For example, wild chimpanzees exposed to naturally occurring vocalisations of conspecifics during feeding demonstrated a significant nasal dip [52–56]. The nasal dip was stronger when meat was available as compared with figs. The authors suggest that the presence of meat increased stress states because it is a more precious commodity. One potential theory underlying the mechanistic underpinning of the nasal dip is that it is present in all primates because our brains and bodies evolved to respond to external stressors by increasing sensory vigilance. When stressed, our autonomic nervous system increases sensory acuity by redirecting blood towards the sensory organs for heightened acuity [57]. The movement of blood towards the eyes and ears causes vasoconstriction around the nose, in turn creating a marked temperature drop in the nasal tip, compared with no acute stressor.

The studies reviewed here indicate that IRT is an emerging, powerful, unobtrusive and ecologically valid methodology to investigate physiological markers of stress and their associations with psychological stress. However, we have yet to demonstrate general population ranges of nasal temperature fluctuations associated with stress, or recovery rates following stress exposure (e.g., return to homeostasis and associations with perceived stress). Much like heart rate variability [58] where the body continuously seeks to maintain homeostasis [59], the nasal dip recovery rate may offer non-contact insights into individual differences in stress regulation. The present study aims to extend our understanding of the human stress response using real-life active stress scenarios and non-contact, continuous IRT video capture to reveal the ranges of nasal temperature fluctuations in the general population associated with psychological stressors and associations with perceived stress.

## Methods and materials

All methods were performed in accordance with the relevant guidelines and regulations of the 1964 Declaration of Helsinki. Ethical approval for the current study was authorised by the University of Sussex's Committee for Science and Technology (ER/MP766/2).

### Participants

Twenty-nine adults (19 females, 10 males) between the ages of 19–43 years (mean = 24.66 ± 6.02) participated in the study. Participants were recruited between 01/10/2024 and 17/12/2024. Exclusion criteria included circulatory conditions, diseases that could influence vasomotor activity [60–61] consumption of vasoactive or psychotropic medications on the day of testing [61] or taking vasoactive substances (e.g., caffeine, alcohol) in the three hours preceding testing. No thermal obstructions were allowed during the data collection (e.g., glasses). Participant numbers in existing literature investigating nasal dips during stress conditions range from 12–28 (mean = 20) [3,23,47]. This average was considered and exceeded when designing the study. A post hoc power analyses (see Results) confirmed our sample size was sufficient to detect expected effects.

### Procedure

Participants were recruited using digital and paper flyers around the university and through the School of Psychology's research participation system. They were informed that the study involved being filmed with a thermal camera during a stress-inducing task but were given no further details on the type of task(s) they would complete or any specific procedures to avoid expectation effects. All experiments were run between 9.30 am and 2 pm to control for the natural variation

in cortisol levels across the day [62–63]. Participants were positioned at a fixed position facing a table for the duration of the experiment. A thermal camera (Ultra HD 2.7K & 36MP & 24FPS) and a RGB camera were used to record thermal fluctuations and participant identification respectively. Cameras were installed on a tripod side-by-side at eye level and adjusted based on participant height at a distance of one metre from the individual's face.

Upon entering the laboratory, participants underwent a 15-minute habituation period [21,23,34] to acclimate skin temperature to the ambient room temperature. This period was followed by a white noise baseline period, two counter-balanced stress induction conditions and a white noise recovery period. During the habituation period, participants completed a demographic questionnaire and perceived stress survey. Surveys were named generically (e.g., Survey 1) rather than descriptively. At the end of the study, participants completed a perceived cognitive and somatic anxiety survey, basing their answers on the most stressful point in the study. They were also asked to state which task they found more stressful (Fig 1).

## Surveys

A Demographic Questionnaire required participants to report their age, ethnographic background and sex.

The Perceived Stress Scale (PSS) [5] was employed to evaluate stable stress characteristics of participants. The PSS addresses how different situations affect our arousal levels within a recent timeframe (e.g., one month). It is an effective questionnaire for capturing relatively stable characteristics of how individuals perceive stressful situations in their lives [4]. The PSS is a 10-item self-report questionnaire that measures how stressed the participant has been feeling in the past month; high scores are associated with diagnoses of depression and anxiety [5,64–66]. To calculate a PSS score, a total score is obtained by summing across all items, except for positively stated items (4, 5, 7 and 8) where the score scale is reversed (i.e., 0 => 4; 1 => 3; 2 => 2; 3 => 1; 4 => 0). Higher scores indicate higher levels of perceived stress.

The State-Trait Inventory for Cognitive and Somatic Anxiety (STICSA) was employed following the stressor conditions as a self-report tool of anxiety traits. The STICSA is a 21-item questionnaire aiming to understand both cognitive and somatic anxiety [6]. Participants were asked to base their answers on the most stressful point in the study. Scores are calculated by summing the positive answers and the questions can be divided into two domains (somatic anxiety and cognitive anxiety) and a total score. Higher scores indicate higher anxiety traits.

## Camera specifications

Thermal data was recorded using a FLIR T450sc infrared camera (resolution: 320 x 240 pixels, frame rate: 30 Hz). The camera operates in the long-wave infrared range (7.5–13.0 um) and has a thermal sensitivity of <0.045°C at 30°C (NETD <45 mK). A fixed emissivity value of 0.98 was used to approximate human skin [67–68].

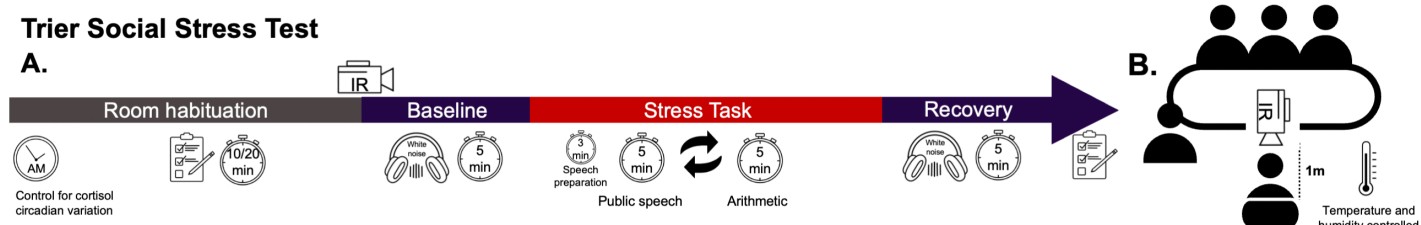

**Fig 1. Experimental Protocol: (A) During the habituation period, participants completed the Perceived Stress Scale (PSS) and a brief demographics survey, followed by two stressor conditions.** After a recovery period, participants completed the State-Trait Inventory for Cognitive and Somatic Anxiety (STICSA). **B.** Room configuration during data collection: participants were seated one metre from the thermal camera, facing a panel of three assessors in interview style. Room temperature and humidity were monitored.

## Environmental measurements

The room temperature and relative humidity were monitored after a 15-minute acclimatisation period before each session began (mean temperature = 23.14°C ± 1.65°C; mean humidity = 42.41% ± 4.80%) [23,34] using a digital thermometer/hygrometer. Camera settings for ambient temperature and humidity (as measured by a thermometer and hygrometer) were set at the start of the experiment. They were rechecked following the room habituation period to ensure accurate thermal readings were acquired. These factors influence how infrared energy travels from the target to the camera. Without correction, changes in ambient temperature and humidity can distort the temperature measurements.

## Stressor conditions

An adapted Trier Stress Test (TSST) protocol was employed in this study because the TSST stands out as an established and validated method to systematically induce psychological and associated physiological stress responses in participants within a naturalistic setting [69–71]. The TSST generally consists of a speech performance (social stressor) and a verbal mental maths test (cognitive stressor), followed by a recovery period [72]. A meta-analysis reports that the TSST is the most sensitive test of psychological stress and is closely associated with the body's physiological stress response [73]. Exposure to the psychological stress conditions of the TSST is correlated with increases in heart rate, blood pressure and several stress markers including the hormone cortisol [74]. The TSST consistently reports statistically significant associations between psychological and physical stress measures in healthy volunteers and clinical populations during real-life behaviour, making the TSST unique for its ecologically grounding and reliability [61]. To date, the TSST has proved effective for eliciting psychological stress (associated with physiological stress) in children, adolescents and adults; however, stress responses can be moderated by factors including age, gender and clinical diagnosis [75].

Following a baseline session, the standard TSST protocol was implemented, consisting of a 5-minute speech task and a 5-minute mental arithmetic task. Although both conditions intend to elicit psychological stress [76], to control for the effect of task order on IRT stress responses, the speech and arithmetic tasks were counterbalanced across participants and all interaction with the participant was scripted (S1 File). After completing both stress induction conditions, participants underwent a white noise recovery period. This sequence is illustrated in Fig 1.

## Thermal baseline and recovery

Both before and after the TSST stressor conditions, participants completed white noise relaxation session. In both periods (pre-test baseline and recovery), participants listened to white noise through headphones for five minutes (volume self-adjusted). This auditory stimulus was used to promote a neutral auditory environment and reduce stress, helping to establish a physiological baseline [77]. During these sessions, researchers present left the room, and participants were encouraged to relax and close their eyes. While findings are mixed regarding the anxiety-reducing effects of white noise [78–79], it has been found to be effective in improving sleep and anxiety in some contexts [80–81]. The recovery session provided a post-test baseline from which thermal recovery rates could be calculated.

## Thermal data processing

FLIR Research Studio (© Teledyne FLIR LLC) was used to analyse thermal recordings. Participants' nasal skin temperature was extracted at a sampling rate of once per second. For a subset of participants (n = 16), high-frequency sampling (1 Hz) was conducted throughout the whole experimental timeline. For another subset (n = 13), low-frequency sampling (1 frame every 2 minutes) was used to identify targeted thermal measures, with high-frequency sampling performed around these specific points. Method reliability tests revealed a strong agreement across the two methods (S1 Table). To avoid augmentation of nasal temperatures only forward-facing frames were sampled, where both eyes of the subject were clearly visible (maximum angle of 45°) [52,53,82,83]. Frames in which the facial image was blurred, or the nasal region

was obstructed were excluded from analysis. Participants were asked to avoid excessive movement or touching their face. Where participant touched their face, a ten second sampling pause was initiated to mitigate against warming or cooling due to touch. Based on this sampling protocol, fewer than 10% of sampled frames were discarded.

To derive nasal temperatures, a Region of Interest (ROI) was manually drawn as an ellipse using standardised anatomical landmarks applied to each participant's nasal region (Fig 2.). The Nose tip ROI was drawn from the centre of the nose to the nose tip. As air flow entering and exiting the nose can create artefacts in temperature measurements, The ROI avoided the nostrils. We extracted minimum, maximum, and mean temperatures during experimental conditions. For analysis, mean temperatures of the nose tip were used. All nasal ROI samples included >50 pixels. Data were extracted from the beginning of the first set of white noise and ended at the end of the second phase of white noise. The thermal analysis was carried out by four coders trained by a researcher, who was experienced with extracting thermal data. Inter-coder reliability was performed on n = 139 images, showing a strong degree of agreement (Cohen's Kappa test are as follows: Nose tip: ICC3 = 0.99 p < .001).

### Planned analyses

The following variables and statistical tests were employed to assess predictions. 1) A significant nasal dip will be elicited in both stress conditions, compared with baseline maximum nasal temperatures during white noise conditions. Wilcoxon signed-rank tests are employed due to violations of normality for most variables (Shapiro-Wilk p < .001 for baseline and recovery); 2) the speech stress condition will elicit a stronger thermographic stress response compared with the arithmetic condition using Wilcoxon signed-rank tests. Wilcoxon signed-rank tests are employed due to violations of normality for

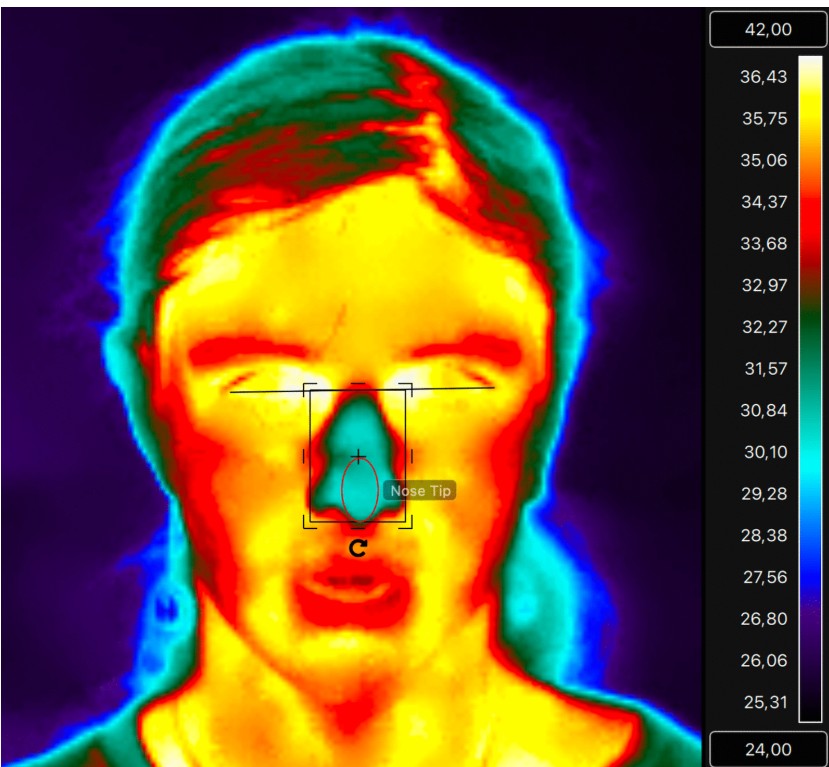

**Fig 2. Nasal Tip ROI.** This image shows the derivation process for the Nasal Tip ROI used to calculate mean temperature. Ellipses were used to avoid nostril area (prevent air flow artefact).

most variables (Shapiro-Wilk p < .001 for baseline and recovery); 3) Surveys for perceived stress and anxiety traits associate with IRT stress responses and recovery rates. Partial Spearman correlations were computed between nose thermal variables and psychometric scores, controlling for ambient temperature and age. To control for the false discovery rate across multiple comparisons, p-values were adjusted using the Holm-Bonferroni method. Results were considered statistically significant when the adjusted p-value was below the α threshold of .05.

It should be noted that IRT responses are known to be sensitive to ambient temperature [36], while TSST protocols report sensitivity to age [75]. Our results revealed significant positive correlations between IRT responses and room temperature. In contrast, age was negatively correlated with a series of IRT responses. Therefore, analyses between IRT responses and survey scores controlled for room temperature and age (S2 File; S1 Fig).

## Results

### Statistical power

Post hoc power analyses were performed using a 95% confidence interval, 80% power, and a one-sided hypothesis. They indicate that, for a paired-samples t-test with a large effect (d = 0.8), the required sample size was estimated at n = 11.14. For a one-tailed correlation with a large effect (r = 0.5), the required sample size was n = 22.61. The choice of a large effect size reflects prior findings in comparable studies on nasal temperature changes during stress, while the one-sided hypothesis was justified by our directional prediction based on existing literature. Our sample size exceeded both thresholds, indicating sufficient statistical power for the analysis conducted.

### Descriptive statistics

Table 1 reports descriptive statistics for facial skin temperature measures, recovery indices, environmental conditions, and psychological scores across 29 participants (for individual participant data see S2 Table). Core temperature metrics include maximum temperature during baseline, minimum temperatures during two stress-inducing tasks (speech and

**Table 1. Summary of descriptive statistics for facial skin temperature measures, recovery indices, environmental conditions, and psychological scores across 29 participants.**

| Variable | Mean | SD | SE | n |
|---|---|---|---|---|
| Initial Temperature (°C) | 32.56 | 2.58 | 0.48 | 29 |
| Maximum Baseline Temperature (°C) | 34.41 | 2.21 | 0.41 | 29 |
| Minimum Temperature During Speech Task (°C) | 31.12 | 2.41 | 0.48 | 29 |
| Time to Minimum Temperature – Speech Task (sec) | 153.08 | 111.73 | 27.93 | 16 |
| Temperature Drop During Speech Task (°C) | 3.30 | 1.58 | 0.3 | 29 |
| Minimum Temperature During Arithmetic Task (°C) | 31.89 | 2.38 | 0.44 | 29 |
| Time to Minimum Temperature – Arithmetic Task (sec) | 154.83 | 120.10 | 30.02 | 16 |
| Temperature Drop During Arithmetic Task (°C) | 2.52 | 1.37 | 0.26 | 29 |
| Temperature After 5 Minutes Recovery (°C) | 33.36 | 3.23 | 0.66 | 29 |
| Thermal Recovery Rate at 5 Minutes (%) | 68.63 | 43.53 | 8.08 | 29 |
| Participant Age (years) | 24.66 | 6.02 | 1.12 | 29 |
| Ambient Temperature (°C) | 23.14 | 1.65 | 0.31 | 29 |
| Ambient Humidity (%) | 42.41 | 4.80 | 0.89 | 29 |
| PSS – Total Score | 15.72 | 5.64 | 1.05 | 29 |
| STICSA – Total Score | 46.86 | 14.31 | 2.66 | 29 |
| STICSA – Cognitive Anxiety Subscore | 22.10 | 7.72 | 1.43 | 29 |
| STICSA – Somatic Anxiety Subscore | 24.76 | 8.25 | 1.53 | 29 |

arithmetic), and post-task recovery temperatures. These variables offer insight into thermal stress reactivity and recovery dynamics, and their relationship with individual psychological and environmental factors. The calculations used to derive each variable (S3 File) depicts the continuous nasal temperature fluctuations of a single participant throughout the time-line of the experiment, consisting of 1549 sampled thermal video frames (Fig 3).

### Nasal temperature

**Stress conditions.** Wilcoxon signed-rank tests were performed to assess differences between stressor conditions (speech and arithmetic) with white noise relaxation conditions prior to stressors and after a recovery period. Comparisons included: 1) Maximum Baseline Temperature and Minimum Temperature During Speech Task, 2) Maximum Baseline Temperature and Minimum Temperature During Arithmetic Task, 3) Minimum Temperature During Speech Task and Maximum Temperature After 5 Minutes Recovery, 4) Minimum Temperature During Arithmetic Task and Maximum Temperature After 5 Minutes Recovery, 5) Maximum Baseline Temperature and Temperature After 5 Minutes Recovery.

Nose skin temperature significantly decreased during both the speech task (m = 31.12 °C ± 2.41°C; V = 435, p < .001) and the arithmetic task (m = 31.89°C ± 2.38°C; V = 435, p < .001), compared to pretest baseline levels (mean = 34.41°C, ± 2.21°C). Five minutes post-task, nasal temperatures were significantly higher than both stress tasks (speech: p < .001; arithmetic: p < .001), however they remained significantly lower than the baseline temperature

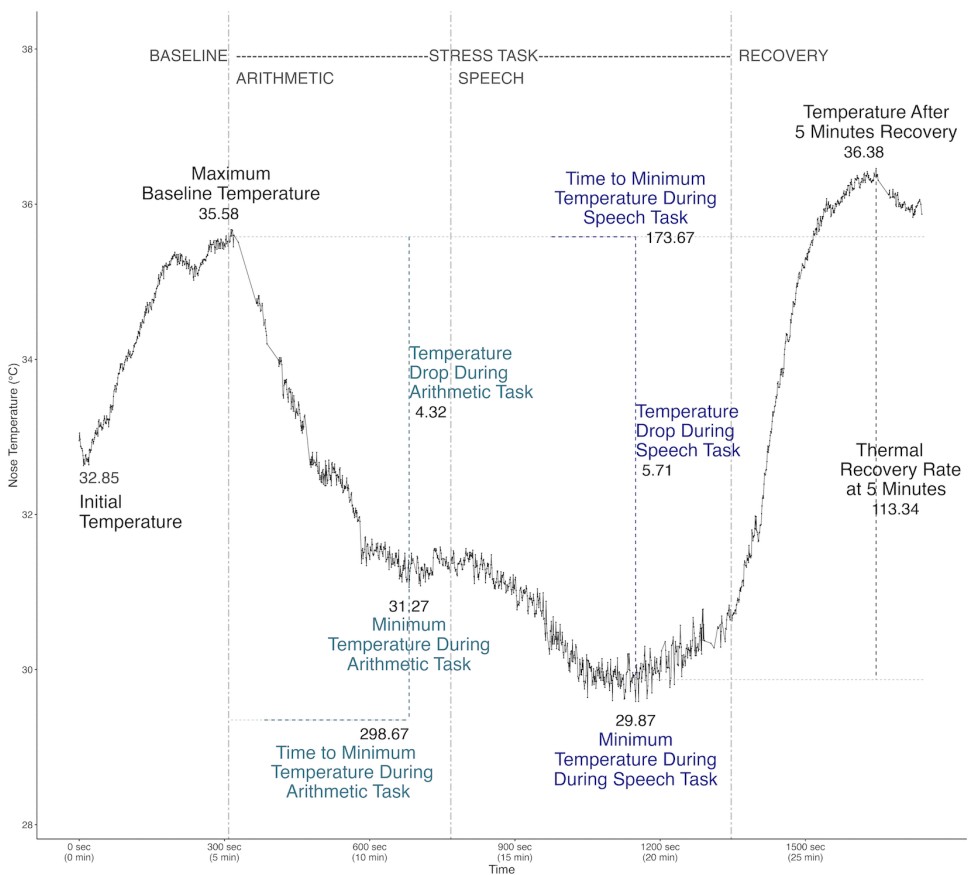

**Fig 3. Continuous nasal temperature variations of a single participant.** Visual representation of nasal temperature fluctuations (consisting of 1549 video frame samples) and variable derivation: baseline and recovery (black), arithmetic stress task (light blue) and speech stress task (dark blue).

(m = 33.36°C ± 3.23°C; V = 352.5, p = .003) A Wilcoxon signed-rank test also revealed that the minimum temperature was significantly lower for the speech task compared with the arithmetic task (V = 68.5, p = .001) (Fig 4).

A Wilcoxon signed-rank test was performed to test the impact of white noise during baseline period. Specifically, this considered the difference between the participant's nasal temperature once habituation to room temperature compared with the maximum temperature during pre-stress test white noise session. Nasal temperature (m = 32.59°C ± 2.58°C) increased significantly (V = 0.84, p < .001) after the white noise period (m = 34.41°C ± 2.21°C) (Fig 5).

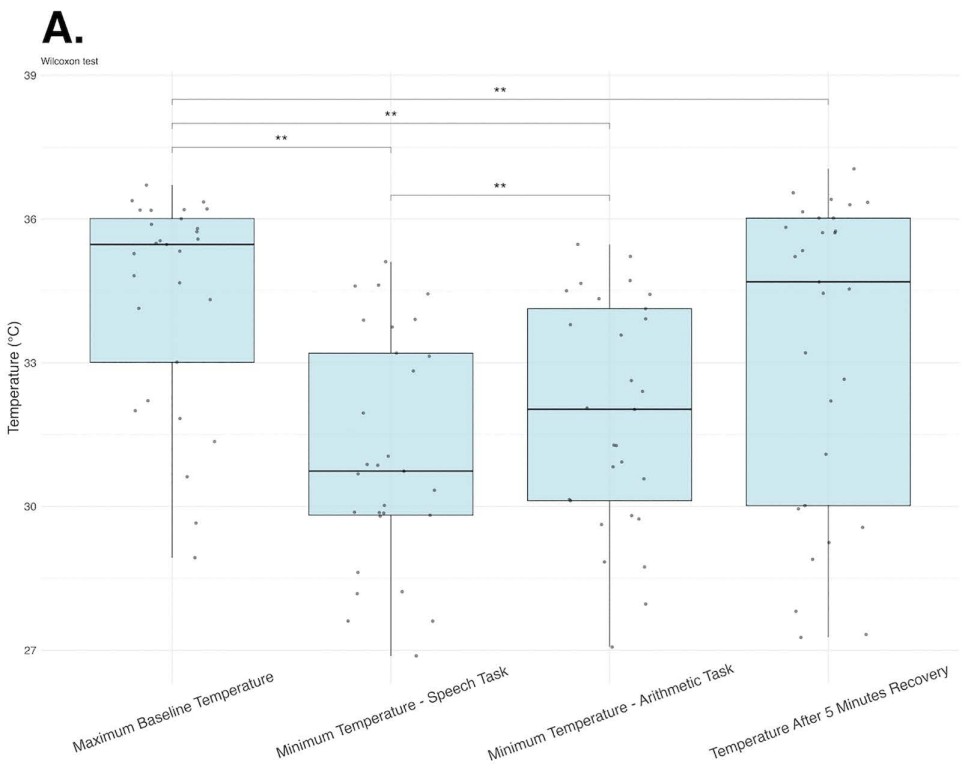

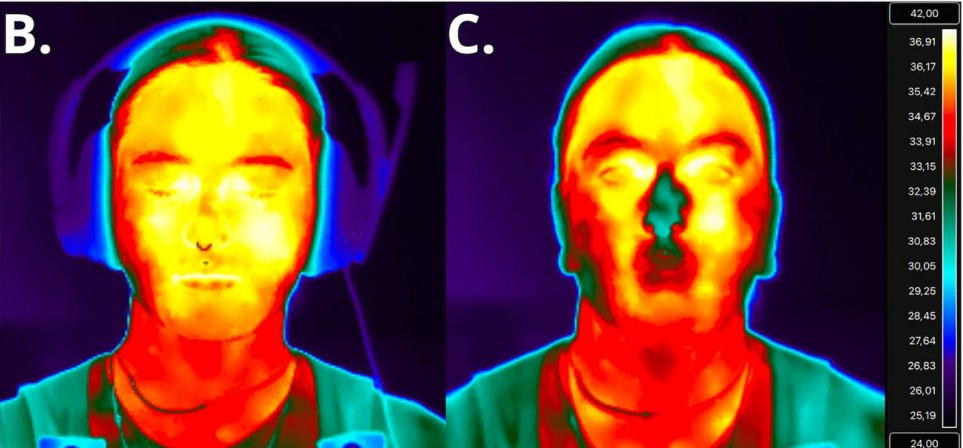

**Fig 4. Nasal dip results.** The top panel (A.) shows mean temperatures across experimental conditions. The lower panel provides an example the nasal dip in a single participant, comparing maximum baseline temperature during white noise condition (B.) and minimum temperature during stressor task (C.) for one participant.

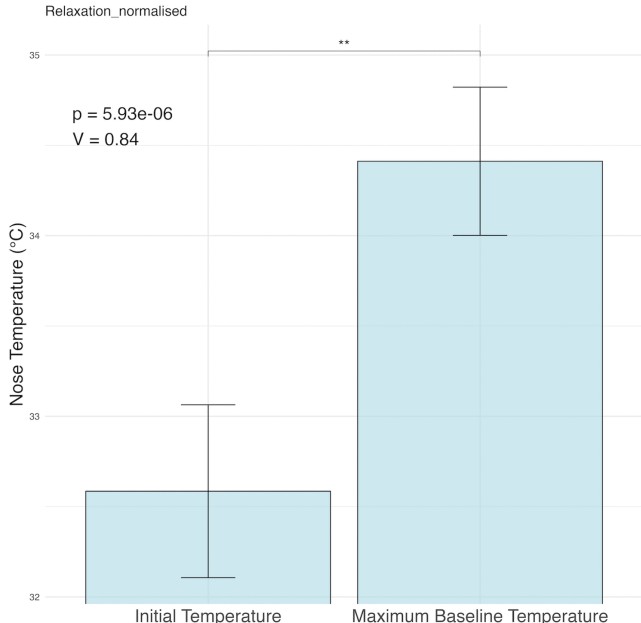

**Fig 5. White noise effect.** Nasal temperatures significantly increased with white noise exposure compared with nasal temperature following room habituation.

**Temporal dynamics of nasal dip.** Among the 29 participants, 16 were coded using an extremely high frequency method. Within this subset of participants, we were able to evaluate the time course of the nasal drop. Seven individuals performed the speech task first and nine individuals performed the arithmetic task first. Regardless of task order, the magnitude of the temperature drop significantly correlates with the time to reach the minimum temperature ($\rho = 0.567$, $p = .02$ tasks collapsed; $\rho = 0.93$, $p < .01$ speech; $\rho = 0.77$, $p = .02$ arithmetic) (Fig 6).

**Ambient temperature and age.** Spearman's rank correlation coefficient tests were run to investigate the impact of age and ambient temperature on nasal skin temperature measures. Several positive associations were found between nasal thermal measures and ambient temperature. Several negative associations were found between age and nasal thermal measures (S2 File; S1 Fig).

**Nasal temperature and perceived stress.** A partial Spearman's rank correlation coefficient, controlling for age and ambient temperature, was performed to assess the link between facial skin temperature and psychological scores. The STICSA – Total Score ($\rho = 0.46$, $p = 0.01$) and the STICSA – Somatic Subscore ($\rho = 0.47$, $p = 0.01$) were positively correlated to the drop in nasal temperature during the speech task (Fig 7; S3 Table for complete results).

**Multiple comparisons.** All comparisons that were significant before correction remained significant after Holm–Bonferroni adjustment. Adjusted p-values ranged from <.01 for all differences between Maxi-mum Baseline Temperature, Minimum Temperatures during the Speech and Arithmetic tasks, Temperature after 5-Minute Recovery versus the minima of both stress conditions, and the Chi-square for the Most Stressful Task, to.01 for the difference between Maximum Baseline Temperature and Temperature after 5-Minute Recovery, and up to.041 for associations between STICSA scores and nasal temperature drop, as well as sex differences in PSS scores.

## Discussion

This investigation extends previous research on stress with a novel experiment that captured physiological stress responses during the administration of an established psychological stress induction protocol. We presented participants

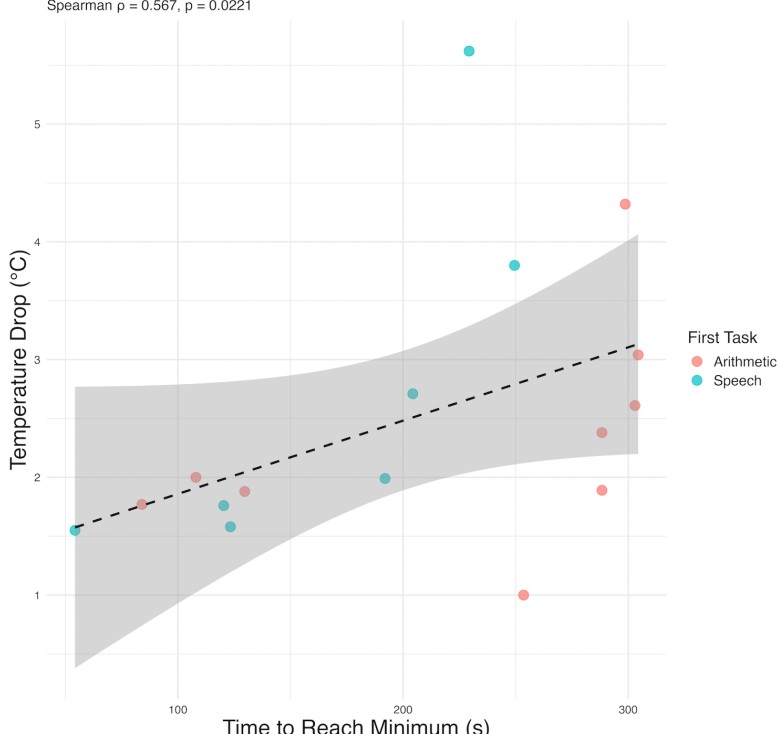

**Fig 6. Time to drop.** Association between timing and magnitude of temperature drop. A significant relationship was observed between the magnitude of the temperature drop and the time it occurred.

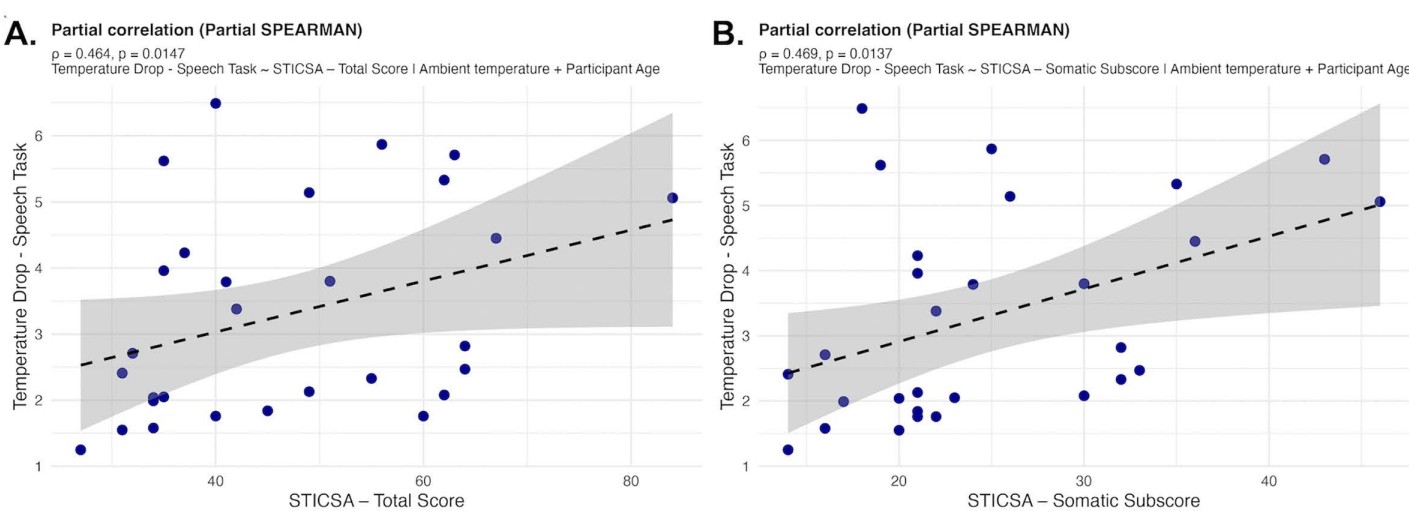

**Fig 7. Nasal Dip and Perceived Anxiety.** STICSA Total Score (A.), driven by Somatic Subscore (B.), correlated with the Temperature Drop during Speech Task.

with an adapted version of the TSST, whilst conducting continuous IRT video recordings of their nasal temperature. Participants also reported stable characteristics of their stress perception (via the PSS) and ratings of their somatic and cognitive anxiety traits in response to the TSST stress induction. The novelty of the study is that it provides continuous, non-contact psychological measures of psychological stress within a real-life scenario.

Consistent with previous research, we found that nasal tip temperature, captured via IRT, is sensitive to negative phycological stressors induced using the TSST speech and arithmetic stress conditions [84]. We report a significant decrease in nasal temperature associated with both the speech and arithmetic conditions compared with individual baselines when controlled for ambient temperature and age. Results were not influenced by sex. The drop in nasal temperature during the stress induction is consistent with sympathetic vasoconstriction, a well-documented physiological mechanism during stress exposure [31]. To add robustness to these results, we report a significant rise in nasal temperature between initial temperature (after room acclimatisation) and maximum baseline during white noise exposure demonstrating the bidirectional change associated with stress and no-stress conditions. We also report that in a subset of our participants (with high frequency sampling) there is an association between the time to reach the minimum temperature (from beginning of the task) and size of temperature drop regardless of the task. Individuals who exhibited a greater thermal response (during induced stress) took longer to reach their minimum nasal temperature compared with those who dropped less. In future studies, the amount of time that elapses to reach one's baseline no-stress and minimum stress-induced nasal temperature may provide a unique temporal window into the workings of the physiological stress system. We intentionally did not conduct contact measures of stress in the present study. However, we predict that biological measures like heart rate, blood pressure, and stress hormones (e.g., cortisol) would rise associated with stress induction and return to baseline post stress, while heart rate variability would rise in association with a body returning to rest. For now, our pattern of results demonstrates both the robustness of the nasal response to the TSST negative stress stimuli and the sensitivity of IRT to capture the stress response. IRT offers a powerful, non-contact opportunity to obtain real-time physiological responses to stress stimuli.

Regarding nasal temperature fluctuations associated with self-reported stress perception, we found the stronger perceived stress condition (counterbalanced for speech and arithmetic stress conditions), as reported by participants, was not associated with their nasal IRT response, although there was an effect of sex. Females found the arithmetic task more stressful while males found the speech task more stressful (S4 File; S2-S3 Figs). For the participants, the speech task elicited a larger decrease in nasal tip temperature compared with the arithmetic stress condition even when participants perceived the arithmetic task as more stressful. From an evolutionary perspective, we speculate that the potential for negative social judgement, is more relevant in the speech task, which produces an acute threat response for social reputation damage [85]. Across primates, damage to social standing is a serious and potentially fatal occurrence. Primate evolutionary theory suggests that social damage can result in group ostracization, which excludes the individual from critical collaborative survival behaviours including the acquisition of food and protection from predation. Therefore, social ostracization can be equated with actual physical demise [86] and still plays out in modern social settings [87–88]. An alternative explanation is that the arithmetic task requires an element of distraction, which has been reported to dampen affective responses to negative conditions including stress [89].

We found no relationships between IRT nasal measures and PSS scores. The PSS is designed to evaluate stress coping mechanisms related to chronic stress and therefore, may not possess the sensitivity for acute stress conditions such as in the present paradigm. On the contrary, we reveal strong significant associations between the IRT stress responses and the STICSA scores, recognised for its ability to differentiate between temporary and chronic anxiety – where feelings of stress are classed as anxiety. The STICSA total and somatic anxiety subdomain scores were positively associated with IRT response to the speech stress task, indicating that the larger the nasal temperature decline, the greater the perceived somatic anxiety. The total STICSA score was primarily driven by the somatic anxiety subscale, supporting the subscale's sensitivity to acute and temporary stressors. These results demonstrate that the perceived stress self-reports reflecting somatic stress sensations, but not cognitive anxiety, aligns with the nasal dip elicited by the same social stressors.

Contrary to expectation, IRT recovery rate (proportional return to baseline) was not associated with the self-reported PSS or the STICSA scores. Although temperatures increased during the recovery phase, they remained significantly below baseline for the population, and in general of recovery rate was highly variable. Some individuals did not recover fully within five minutes while others recovered above their baseline temperature. IRT recovery rates have not been well explored, however, we considered the rate of return to baseline nasal temperature as a measure of resilience and adaptability to ever-changing environments, akin to heart rate variability, which in turn has shown to be negatively associated with perceived stress [14]. Further research is required to better understand both the timeline and the role of IRT recovery associated with psychological stress, perceived stress and anxiety.

This study does not come without limitations. While thermal cameras are widely available as a cost effective, non-contact tool that can be applied in real-life scenarios, sampling continuous video using ROIs remains an onerous endeavour. Without automated ROI tracking to both save time and tackle issues of accuracy in sampling (e.g., from motion), we rely on time-consuming manual identification of facial landmarks with suitable pixel power from frame-to-frame [39–40]. There is also the need to gain additional information on external factors, other than stress-related vasoconstriction, that may influence IRT nasal changes. For example, different physiological functions, like breathing [31] have been reported to induce facial temperature variability that may not always be linear. Moreover, it is not yet clear if IRT can distinguish the affective valence of a stressor. In this study we focused on social stress which is often considered negative. But the act itself is neither positive nor negatively valanced, rather it is the individual's interpretation that associates affect. This is because arousal sits at the core of a single stress system, void of valence. How stress is interpreted as either positive or negative affect, through our physiological and psychological experiences, requires further consideration, particularly if we aim to apply this method to non-verbal populations like infants and non-human great apes.

Regardless of any methodological shortcomings in the wider field, the findings reported here rely on a tightly constrained experimental protocol and rigorous manual sampling to accurately isolate nasal ROIs that are temporally aligned with stressor stimuli. The findings demonstrate robust nasal temperature fluctuations associated with stress and no-stress conditions and also with perceived somatic anxiety. While our participant group is relatively small and will undoubtably possess inherent biases, it is nevertheless an exciting new discovery with significant implications for investigating stress responses in wider populations. Moreover, it could offer potential for developing new applications for screening, diagnosis and interventions in individuals with stress-related conditions across the lifespan and regardless of language capacity. It may also help us to better understand ourselves via comparative studies of human and non-human primates, since we share an evolutionary adaptive physiological response to stressful stimuli. On the other hand, comparative research may also provide valuable information about how stress impacts non-human great apes with implications for positive conservation action. For example, sanctuaries are mainly limited to observed, non-verbal behaviour to make decisions about ape introductions, groupings and within-species adoptions. Because all primates are prone to behavioural masking, the stress information provided via IRT may provide valuable psychological wellbeing information, beyond the mask, helping to increase the success rates of these rehabilitation activities.

To obtain a comprehensive understanding of how physiological and psychological stress associate with each other, and to inform our understanding of the role of stress in wellbeing, development and the evolution, we require future investigations to extend populations to include individuals from infancy through the lifespan, across contexts and species. It is particularly important to extend to different populations, especially where any 'human' norm, or species comparison is being made. In addition, there is a need for cross-population validation and species-specific calibration before broader comparisons can be drawn.

## Supporting information

**S1 File. Protocol script.**
(DOCX)

**S1 Table. Method reliability tests.**
(DOCX)

**S1 Fig. Correlation between ambient temperature and age on nasal skin temperatures measures.**
(TIF)

**S2 File. Ambient temperature and age effect.**
(DOCX)

**S2 Table. Individual participants temperatures for baselines and stress conditions.**
(DOCX)

**S3 Table. Correlations between temperature and psychological scores controlling for age and ambient temperature.**
(DOCX)

**S3 File. Calculations to derive variables for the analysis.**
(DOCX)

**S4 File. Sex and perceived stress.**
(DOCX)

**S2 Fig. Sex effect on PSS Total Score (A) and on self-reported most stressful task (B).**
(TIF)

**S3 Fig. Distribution of tasks with strongest temperature drop (A), accuracy in identifying the most stressful task (B) self-reported task choice and accuracy by sex and most stressful task (C).**
(TIF)

**S5 File. Associations between psychological tests.**
(DOCX)

**S4 Fig. Correlation matrix across psychological tests.**
(TIF)

## Acknowledgments

Thank you to the participants who provided their time and their stress in the name of science.

## Author contributions

**Conceptualization:** Perrine Theroude, Marianne Paisley, Georgina Donati, Gillian S. Forrester.

**Data curation:** Perrine Theroude, Marianne Paisley, Matthew Thompson, Amelie Wheeler, Gillian S. Forrester.

**Formal analysis:** Perrine Theroude, Marianne Paisley, Gillian S. Forrester.

**Funding acquisition:** Gillian S. Forrester.

**Investigation:** Perrine Theroude, Marianne Paisley, Matthew Thompson, Amelie Wheeler, Gillian S. Forrester.

**Methodology:** Perrine Theroude, Marianne Paisley, Matthew Thompson, Amelie Wheeler, Georgina Donati, Gillian S. Forrester.

**Project administration:** Gillian S. Forrester.

**Supervision:** Georgina Donati, Gillian S. Forrester.

**Writing – original draft:** Perrine Theroude, Marianne Paisley, Gillian S. Forrester.

**Writing – review & editing:** Perrine Theroude, Marianne Paisley, Georgina Donati, Gillian S. Forrester.

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
