## [Decision Letter · Decision Letter 0]

23 Oct 2025

Dear Dr. Forrester,

Thank you for submitting your manuscript to PLOS ONE. After careful consideration, we feel that it has merit but does not fully meet PLOS ONE’s publication criteria as it currently stands. Therefore, we invite you to submit a revised version of the manuscript that addresses the points raised during the review process.

We look forward to receiving your revised manuscript.

Kind regards,

Giulia Prete

Academic Editor

PLOS ONE

Journal Requirements:

3. We note that Figure 1 in your submission contain copyrighted images. All PLOS content is published under the Creative Commons Attribution License (CC BY 4.0), which means that the manuscript, images, and Supporting Information files will be freely available online, and any third party is permitted to access, download, copy, distribute, and use these materials in any way, even commercially, with proper attribution. For more information, see our copyright guidelines: http://journals.plos.org/plosone/s/licenses-and-copyright.

Reviewers' comments:

Reviewer's Responses to Questions

**Comments to the Author**

1. Is the manuscript technically sound, and do the data support the conclusions?

Reviewer #1: Yes

Reviewer #2: Yes

2. Has the statistical analysis been performed appropriately and rigorously?

Reviewer #1: Yes

Reviewer #2: Yes

3. Have the authors made all data underlying the findings in their manuscript fully available?

Reviewer #1: Yes

Reviewer #2: Yes

4. Is the manuscript presented in an intelligible fashion and written in standard English?

Reviewer #1: Yes

Reviewer #2: Yes

Reviewer #1: This is a well written manuscript. The study findings are sound. Investigating Psychological Stress with Thermal Imaging” represents an important and timely contribution to the field of stress research. Stress remains one of the most pressing challenges to public health, yet many of the tools available for its measurement rely heavily on subjective reports or invasive procedures, both of which have limitations in ecological validity and applicability. By introducing a non-contact, continuous, and objective method through thermal imaging, the authors successfully advance a new frontier in stress assessment.

One of the most compelling strengths of this work is its innovative use of thermal video to track nasal temperature fluctuations as a proxy for stress. The study not only demonstrates that induced stress leads to measurable decreases in nasal temperature, but also distinguishes between different forms of stress: social stress producing a more pronounced effect than cognitive stress. This differentiation underscores the sensitivity and potential granularity of thermal imaging as a physiological marker of stress.

Another notable finding lies in the nuanced analysis of subjective and physiological correlates. While perceived stress did not align directly with nasal temperature changes, the link between somatic anxiety symptoms and thermal fluctuations provides an intriguing window into how the body encodes and expresses psychological strain. This finding alone invites further exploration and could inform both basic research and clinical practice.

The methodological rigor of the study, combined with its forward-looking implications, makes it highly relevant across a wide range of domains—from clinical psychiatry and psychology to occupational health, education, and even non-verbal populations where traditional stress assessments are impractical. In highlighting the translational value of thermal imaging, the authors pave the way for real-world applications that could reshape stress detection and management.

Overall, this is a fine and highly promising study, offering originality, methodological sophistication, and genuine clinical and societal relevance.

Reviewer #2: This is a very nice addition to the developing literature on the use of thermal measurements as indicators of stress. It is largely clear and well written, and makes an important contribution in establishing more fine-grained and dynamic measurements—the benefits of which are highlighted by the between participant variation in timescales of response.

My comments below are generally just for clarity, or suggestions of where the ms would benefit from a little more detail.

Specific comments

Abstract:

Some technical terms in here that aren’t immediately clear to the naive/general reader (e.g., I’m not sure what a ‘contact measure’ is in this context, why is ‘non-contact’ advantageous). Might be useful to clarify.

Line 42,44, etc: I’m a fan of the Oxford commas in lists to help the reader parse meaning and there are examples throughout the abstract and introduction where they would be useful in correctly interpreting items vs combinations on a list.

Introduction:

A general read over for text smoothness/punctuation might be useful to ease things for the reader (e.g., perceive we are threatened (line 60), previously stressful situations (line 62), and comma/clause use throughout).Line 88: suggest ‘these more objective stress measures’

Line 122: suggest ‘stress induced by using fear’

general - there is a general framing of stress as a negative experience - whereas it can be both negatively and positively valenced (e.g., excitement, arousal). I wonder if it’s useful to briefly address this somewhere, before concentrating on the negative/fear side of things.

Line 134: typo ‘decreases are directly associated’

Line 150: typo ‘a relatively low sample rate’

Line 166: typo ‘meat was available as compared to figs’

Line 170: there’s something about the grammar here (sensorily vigilant..? show sensory vigilance..?).

Line 172: typo, -which (edit to comma)

Methods

Line 194: typo ‘females’

Again - the text would benefit from consistent use of punctuation to ease the reading experience

It might be helpful to have some brief details on recruitment that clarify what the participants knew about the test and reasons for testing (given the white coat effect mentioned earlier in the intro may well extend to participation in a study).

Line 260: in what ways were the camera settings adjusted and why (was this to set a different baseline?, window of sensitivity?). I’m not an expert here, and so it would be useful for replication to know more, and to have a sense of why this was needed (rather than controlling for variation in a statistical model for example).

Line 265: typo ’to systematically induce..’

Line 304: typo ‘were sampled’

Line 308: typo ‘based on this sampling protocol’

Discussion

Line 475: typo ‘individual baselines’

Line 481: typo ‘participants’

Lines 483-486: can the authors draw any parallels with other physiological processes here that would connect to the wider literature on human body processes/health e.g., time to heart rate baseline indicating cardiac fitness, or hormonal changes in response to stress, etc. (I see it’s briefly brought in later (~line 527) - but this could be expanded on/introduced here).

Again, punctuation use throughout could benefit from a thorough tidy over.

Line 496: I’d typically refer to a group this small as ‘for the participants’ rather than ‘population’

Line 500-506. ‘Ancient man’ is a slightly odd/outdated term and I’d suggest an alternative. But you could just refer to ‘primates’ in general (given that we have no direct evidence of whether and how our ancestors used violence in response to social standing). Could you provide a little more detail or support for the idea that ‘social damage can result in group ostracization’ - I don’t doubt that it may in some species in some circumstances, but I’m not sure how much actual evidence we have of this - nor how generalisable that is across primates (and in cases the opposite has been shown).

Line 521: there’s an error in sentence structure in here somewhere, but I’m not sure what to suggest to fix it.

Line 531-543: while the sample was sufficiently powered I assume that it was systematically biased in other ways in terms of trying to describe ’human’ level physiological responses? These may be particularly important to consider where there are suggestions being made about ‘typical’ baselines/responses/therapeutic use.

Line 555: there is a teaser here about implications for positive conservation action, that would be interesting to unpack (I can more clearly see the welfare implications for captive living animals, but these could be clarified too?).

Line 560: as above I think it would be particularly important to note the need for extension to different populations, especially where any ‘human’ norm, or species comparison is being made.Figure 4: the box plots with jitter are very nice, but perhaps it would be useful - given this is a within subjects test, to show the slopes of the lines between the points for each participant. It’s clear there’s a lot of between subject variation - but it would be nice to know if this extended to the slope between conditions as well as within them. This may be too small a sample to do so statistically - but a visualisation of the data would be a nice addition.

**Do you want your identity to be public for this peer review?** For information about this choice, including consent withdrawal, please see our Privacy Policy

Reviewer #1: **Yes: ** George P. Chrousos

Reviewer #2: No

---

## [Author Response · Author response to Decision Letter 1]

17 Nov 2025

We thank the Editor and Reviewers for their thoughtful comments. We are pleased now to submit a revised manuscript.

---

## [Editor Report · Decision Letter 1]

19 Nov 2025

The Nose Knows: Thermal Responses to Active Psychological Stressors

PONE-D-25-45280R1

Dear Dr. Forrester,

We’re pleased to inform you that your manuscript has been judged scientifically suitable for publication and will be formally accepted for publication once it meets all outstanding technical requirements.

Kind regards,

Giulia Prete

Academic Editor

PLOS ONE

---

## [Editor Report · Acceptance letter]

PONE-D-25-45280R1

PLOS One

Dear Dr. Forrester,

I'm pleased to inform you that your manuscript has been deemed suitable for publication in PLOS One. Congratulations! Your manuscript is now being handed over to our production team.

Kind regards,

on behalf of

Dr. Giulia Prete

Academic Editor

PLOS One